# Micro Light Flow Controller on a Programmable Waveguide Engine

**DOI:** 10.3390/mi13111990

**Published:** 2022-11-16

**Authors:** Tao Chen, Zhangqi Dang, Zeyu Deng, Zhenming Ding, Ziyang Zhang

**Affiliations:** 1College of Information Science and Electronic Engineering, Zhejiang University, Hangzhou 310027, China; 2Laboratory of Photonic Integration, School of Engineering, Westlake University, Hangzhou 310024, China

**Keywords:** programmable photonic integrated circuits, arbitrary power splitter, optimized algorithm, thermo-optic effects

## Abstract

A light flow controller that can regulate the three-port optical power in both lossless and lossy modus is realized on a programmable multimode waveguide engine. The microheaters on the waveguide chip mimic the tunable “pixels” that can continuously adjust the local refractive index. Compared to the conventional method where the tuning takes place only on single-mode waveguides, the proposed structure is more compact and requires less electrodes. The local index changes in a multimode waveguide can alter the mode numbers, field distribution, and propagation constants of each individual mode, all of which can alter the multimode interference pattern significantly. However, these changes are mostly complex and not governed by analytical equations as in the single-mode case. Though numerical simulations can be performed to predict the device response, the thermal and electromagnetic computing involved is mostly time-consuming. Here, a multi-level search program is developed based on experiments only. It can reach a target output in real time by adjusting the microheaters collectively and iteratively. It can also jump over local optima and further improve the cost function on a global level. With only a simple waveguide structure and four microheaters, light can be routed freely into any of the three output ports with arbitrary power ratios, with and without extra attenuation. This work may trigger new ideas in developing compact and efficient photonic integrated devices for applications in optical communication and computing.

## 1. Introduction

Reconfigured and programmable photonic integrated circuits (PICs) are dedicated to discovering the general on-chip hardware configurations for versatile functions by customized programs [1,2]. Their network architectures are commonly formed by the cascaded Mach–Zehnder interferometers (MZIs), directional couplers (DCs), and micro ring resonators (MRRs), which can be programmed in real time through a series of optical phase shifters applied on the single-mode waveguide arms. Their cascaded mesh connectivity can determine the path of light routing, define the transmission matrix, and perform spectral filtering operations.

On the single-mode waveguide arms, the change of refractive index is translated to the optical phase change only. The cascaded systems can be well-represented by a set of analytical equations or a linear transfer matrix, and the required settings to reach the target output can then be readily calculated. However, there remain two problems. For one, the cascaded structure requires a large number of independent tuning units, which may pose a challenge for electronic wiring during integration and assembly [3,4]. Secondly, the phase tuning on a single-mode waveguide cannot efficiently introduce variable attenuation of light amplitude. In fact, the MZI and DC-based solutions all work under the theoretically lossless condition [5], apart from the insertion loss in practice, i.e., if the power of one output is tuned down, the sum of the output power from the remaining ports is bound to go up.

To illustrate the second problem, let us consider the integrated power splitters, a fundamental PIC component that divides the input optical power into multiple streams. They can be realized by Y-branch structures [6], directional couplers (DC) [7], and QR code-like nanostructure [8]. However, they often perform a fixed splitting ratio, as shown in Figure 1a. The tunable power splitters can be realized by the cascaded MZI-based device [9], as shown in Figure 1b. By precisely tuning each phase shifter, the input light can be divided into arbitrary splitting ratios at the outputs. Note that in Figure 1b, the phase tuning does not affect the overall power passing through the device. To implement extra attenuation, each output port needs to connect to a variable optical attenuator (VOA) [10,11]. The extra VOAs require more independent tuning elements plus methods to suppress the crosstalk among them, thereby increasing the device footprint and cost.

Multimode interference (MMI) devices have been long implemented as compact power splitters [12]. However, most MMIs are used as fixed power splitters, e.g., the 3 dB coupler in a MZI network [13]. Recently, thermally tunable MMIs have been demonstrated as programmable multi-functional PICs [14,15,16]. For example, switching networks of large port counts can be realized with a minimal number of control units in a compact, non-cascaded structure [14]. As shown in Figure 1c, such device consists of only one multimode waveguide with a set of parallel microheater electrodes and input/output access waveguides. However, so far, only the “digital” switching function is achieved [14]. Light can only go from zero to its respective maximum at a given logic output and cannot be tuned freely at any value in between.

The problem lies in the complexity of index tuning in the multimode waveguide. Unlike in the single-mode system, where only the optical phase changes, the tuned local refractive index in a multimode waveguide can simultaneously influence the multiple guided modes, including their total number, field profiles, and propagation constants, all of which can alter the multimode interference patterns as well as the coupling efficiencies to the output waveguides. Thus, under the multimode regime, the outputs cannot be calculated using simple, analytical equations. One can rely on numerical simulations to find the solution, e.g., translating the thermal power to the temperature gradient and further to index profile, obtaining the parameters for each supported mode, adding their respective phase delay over a propagation distance, and finally, calculating the combine optical fields at the output ports. However, such simulation would take a long time, and a substantial number of iterations are needed to sweep the parameters for a reasonably optimal output. This “forward” design method based on simulations is extremely time-consuming. As a result, only a small number of “digital” switching functions were demonstrated in [14].

On the other hand, we believe that local index tuning on a multimode waveguide is a powerful tool, as it can simultaneously alter several factors of a light wave, steer the light interference in the confined space, and in turn dramatically change the amount of light flow at the output ports. Programmable photonic devices can be built with simpler structures, fewer microheaters, and more compact footprints. Therefore, we established a platform called function programmable waveguide engine (FPWE), on which we can apply a set of thermal powers to a MMI and obtain the optical readout directly from experiment without running tedious simulations, i.e., the FPWE itself works as a practical simulator [15,16]. Though similar engines can be built on other material platforms, such as the silicon multimode waveguide with embedded phase change material [17,18], the polymer platform is attractive, as the fabrication process is simple and low-cost, the integration technology with other materials/components is flexible [19,20], and multi-layer waveguide structures can be integrated [20,21], all of which can be well-managed within one small-sized laboratory.

In this work, we explored the FPWE system further and developed a truly flexible light flow controller beyond digital switching and with integrated VOA function. Some preliminary simulations were needed to confirm the size of the multimode waveguide and the microheaters under the constraints of the fabrication technology. The MMI-based chip was directly measured and defined in experiment. Understanding the complex nature of multimode tuning, an “online” search program was developed to drive the chip and collect the feedback like “instant messaging” between input and output. The results of each iteration were instantly analyzed to guide the setting of the next driver update. A levelled searching algorithm was implemented to jump over traps of local optima. We stress that the tuning element/electrodes on the multimode waveguide work collectively, i.e., they do not need to be isolated. The unwanted tuning crosstalk in the single-mode case is in fact a powerful tool in the multimode waveguide. Live light flow switching is demonstrated under various lossless and lossy targets (see Appendix A). We hope this work may trigger interesting applications of multimode waveguide devices for advanced photonic applications.

## 2. Materials and Methods

### 2.1. Waveguide Design and Simulation

The design of the buried multimode waveguide is shown in Figure 2a. Commercial polymer materials from ChemOptics (Daejeon, Korea) are used for the core (ZPU-470) and cladding (ZPU-450). The refractive index of the core and cladding is 1.47 and 1.45 at 1550 nm wavelength, respectively. The thermo-optic coefficient of the polymer material is measured to be −1.14 × 10^−4^/°C. The multimode waveguide is 70 μm in width and 3.5 μm in thickness. The top and bottom cladding thickness is 6 μm and 15 μm, respectively. Four microheaters (named H_1_–H_4_; each has 8 μm width) are placed symmetrically on the surface of the top cladding. The cross sections of the input and output waveguides are set to be 3.5 × 3.5 μm^2^ for the polarization degenerate single-mode operation. The mode field of the single-mode waveguide, shown in Figure 2b, is calculated by Lumerical MODE software. The schematic of the 1 × 3 light flow controller is shown in Figure 2c. The length of the multimode waveguide is set to 2500 μm, and the device functions as an even 1 × 2 splitter at the initial/unaltered state, as shown in Figure 2d. The microheaters have the same length as the multimode waveguide and are symmetrically placed with 25 μm gaps. The taper structures varying from 3.5 μm to 17.8 μm (or reverse) in width are applied to improve the coupling efficiency between the multimode waveguide and the input/output single-mode waveguides.

The total optical field at the output *E_tot_* of the multimode waveguide can be denoted by
(1)$$E_{\mathrm{tot}}(x,y,z)=\sum_{v=1}^{m}c_{v}\phi_{v}{(y,z)e}^{{-j\beta }_{v}x},$$
where *ϕ_v_(y,z)* is the eigenmode field distribution, and *β_v_* is their respective propagation constant. *c_v_* is the coupling coefficient of the individual eigenmode to the input light field, and *m* is the total number of the guided modes supported by the multimode waveguide [22]. The added heat can alter the local refractive index through thermo-optic effect. Subsequently, *m*, *ϕ_v_(y,z)*, *β_v_*, and *c_v_* all become different and can alter the output optical field *E_tot_* significantly. The left column of Figure 3a–c shows the simulated temperature gradient by Lumerical HEAT when microheaters H_1_ (15 mW) and H_3_ (10 mW) are applied individually and simultaneously. The right column shows the calculated field profiles of the fundamental eigenmode. The results indicate that the joint adjustment of H_1_ and H_3_ is not simply the linear sum of their individual contributions. This is an important feature, meaning that in a multimode system described under Figure 2a,b, the electrodes cannot be swept in sequence in search for the optimal thermal condition for a specific output. Instead, a collective optimization method among all electrodes, or at least a group of electrodes, should be applied.

Figure 4a–d show the simulated output intensity of Port1–Port3 and their total outputs when H_3_ varies from 0 to 25 mW under different values of H_1_. All the intensity values are normalized to the total output intensity of the initial 3 dB splitter (Port1 and Port3) without any heater power applied. The output intensity changes dramatically under the different powers applied on H_1_. This again shows that the microheaters can be well thermally coupled in this application, and their influence on the output optical power should be considered collectively.

The amplitude of light in each output is determined by the mode coupling efficiency between the end-facet of the multimode interference profile and the single-mode profile of the respective output waveguide. High loss/large attenuation is expected when the two profiles mismatch, as shown in Figure 4e for the “lossy” condition. If the light is well-imaged on one or some of the output waveguides, as in the classic MMI application, the output coupling efficiency approaches 100%, and the device works under the “lossless” condition, as shown in Figure 4f. Again, we stress that the lossy and lossless conditions refer to the initial case, in which the pure passive MMI works as a 3 dB splitter to Port1 and Port3, as shown in Figure 2d, when no thermal powers are applied. If the total output power is equal to the initial case, we define it as “lossless”, whereas the device is “lossy” when extra attenuation is exerted by shifting the focus or smearing the light field out for a reduced out-coupling efficiency.

By comparison, the adjustment/phase tuning in the MZIs- and MRRs-based single-mode system is often intentionally independent, where thermal crosstalk is an unwanted effect and should be suppressed. Recent work has also shown that the thermal crosstalk can be canceled using the thermal eigenmode decomposition (TED) method [23], considering that the thermally induced refractive index change only causes a linear phase delay in the single-mode waveguide. In this way, all the units on the cascaded structure can allow individual and sequential adjustment. The control matrix is well-defined and can be described with analytical equations. Thus, the realizable functions, e.g., switching and filtering, can be predicted with straightforward arithmetic calculations. On the contrary, as demonstrated in Figure 2 and Figure 3, the adjustment in the multimode waveguide by the microheaters is “lumped”. This is because the temperature gradients from different microheaters can disturb each other in all the three dimensions (only the YZ plane is shown in Figure 3). Thermal crosstalk is allowed and becomes a tool in the MMI configuration. Furthermore, the added heat can simultaneously influence all the guided eigenmodes and further influence the output field according to Equation (1). Thus, it is difficult to derive analytical equations for the transmission matrices. This also requires the consideration of the collective or group effect from the microheaters in the development of the search program for the target output.

### 2.2. FPWE System and Experimental Setup

After the structural design, the device is fabricated using standard process as described in ref. [14,24]. Only standard contact lithography is used to pattern the waveguides and microheaters on a 4-inch wafer. The diagram of the FPWE system is shown in Figure 5a. A CW laser at 1550 nm wavelength is used in this system. Light is coupled into the chip by a standard single-mode fiber (SSMF) aligned and glued on the chip facet. A polarization controller (PC) is added to determine the polarization, which is selected to be TM in this work. An imaging system is placed on the output facet of the chip, and the output spots can be collected by an IR camera (Bobcat-640, 640 × 512 pixels, 16-bit resolution). A microcontroller unit (MCU) current source (advanced ARM-based 32-bit STM32F730xx family) is used to drive the chip. The output direct current (DC) can vary from 0 to 20 mA with a minimum step of 1 μA. Maximal 16 channels can be applied simultaneously, while only 4 are used in this work. The MCU source and IR camera are synchronously controlled by the MATLAB scripts on the controlled computer. The MMI-based chip is considered as a “black box”: the inputs are the applied currents on the electrodes, and the outputs are the resulting optical power from the waveguides as captured by the IR camera.

The photos of the actual system are shown in Figure 5b–d. The chip is diced from the 4-inch wafer and fixed on the customized T-shaped PCB adapter, including electric pins and interfaces. Wire-bonding technology is developed to connect the pads on the chip to the pins on the PCB adapter. A microscope photo of the chip integrated on the adapter and with the input fiber is shown in Figure 5e. The adapter is then connected with the MCU source by the bus lines. The imaging system captures the light power variations from the output facet. An actual camera shot is shown in Figure 5f, where the inset shows the captured light spots, each contained in a square of 40 × 40 pixels.

For practical applications, FPWE system can be made more compact with an integrated laser diode (LD) as light source and photodetectors (PDs) as monitor diodes. The integration technology with these active components have been well-established on polymer waveguide platform, including butt-joint coupling at the facet and on-chip 45° mirrors [21,25]. The electronic control system can also be replaced by a customized high-speed MCU or FPGA circuits. The search program can then be compiled from a general PC and then burned into the microsystem’s ROM.

### 2.3. Software Driver and Search Program

The driver and optimization program are both written as MATLAB scripts and fed to the FPWE via the central computer. The MCU-based current source is driven by writing and refreshing values into the defined registers in MCU through the modbus protocol by USB cable. The real-time monitoring and image capture of the IR camera are realized by the image acquirement toolbox of MATLAB through the GigE Version protocol through an ethernet cable. More details of the hardware drivers can be found in the software Appendix A.

A search program is developed to find the optimal electric currents under a given target optical output. To evaluate how close the actual results approach the target, we first define a cost function (*CF*) using the following equation:(2)$$CF=\left(\sum_{i=1}^{3}\left(\frac{O_{i\_Tar.}}{I_{Total}{}_{\_Tar.}}\times \left(1-\frac{\left|O_{i\_Tar.}-O_{i\_N}\right|}{O_{i\_Tar.}+O_{i\_N}}\right)\right)\right)\times \left(1-\frac{\left|I_{Total}{}_{\_Tar.}-I_{Total}{}_{\_N}\right|}{I_{Total}{}_{\_Tar.}+I_{Total}{}_{\_N}}\right),$$
where *O_i_* is the power of the respective three output ports of the target (*Tar.*) and from the *N^th^* experimental (*N* is the iteration number). The optical power is calculated by summing up the 40 × 40 pixel counts at the respective output spot. *I_Total_Tar_.* is the total output power of the target. The *CF* in Equation (2) is firstly calculated by summing up all the three power differences by multiplying their proportion in the target total output power. It is then normalized to the range from 0 to 1, by multiplying the difference between the actual total output power *I_Total_N_* and the target total output power *I_Total_Tar_*. *CF* = 1 means that the experimental results match completely with the target, while *CF* = 0 means that they are completely mismatched.

To define the lossless and lossy functions, we first measure the total output power *I_init_* at the initial state in which no heaters are applied, and the device works as an even 3 dB splitter (to Port *O*_1_ and *O*_3_) from the design. The total pixel count (1.02 ×10^7^) of *I_init_* is then set as the reference. The outputs are considered lossless when the total power is equal to *I_init_*, i.e., *O*_1_ + *O*_2_ + *O*_3_
*=* 1, meaning that no extra scattering loss is introduced in the thermal tuning process. On the other hand, the outputs are considered lossy when the total power is intentionally attenuated to a level below *I_init_*, i.e., *O*_1_ + *O*_2_ + *O*_3_
*<* 1.

The flow of the search program is depicted in Figure 6. First, the target total power and the splitting power among the three ports are set. It must be noted that the setting of the target total power determines the lossless or lossy splitting functions. Then, an initialization current configuration (*E_0_*) of H_1_ to H_4_ is generated by a random function (0.00~9.00 mA, which can be flexibly adjusted for different designs) from MATLAB (step **S-0**). Two levels of search are included in this work, and more levels can be added for advanced applications. The level-1 adjustment is used to produce a dramatic change that causes the iteration to leap out of the locally optimized solutions. The level-2 is used to accurately search for the optimized values in the local range. In each iteration, the experimental total power *I_Total_N_* is firstly compared with the target total power *I_Total_Tar_.* (step **S-2**). If the *I_Total_N_* is in the range of the target (±20%, which can be changed or added extra conditions for different designs, e.g., the *CF* should be >0.8 at the same time), the cost functions between the present *CF_N_* and the last *CF*_*N*−1_ iterations are then compared (step **S-3**). Otherwise, the configuration of the four microheaters is reinitialized with a random function ranging from 0.00 to 9.00 mA. In the level-1 adjustment, each microheater has an independent value, and all four microheaters update at the same time.

In the following *CF* comparison, after a matched total output power range is found, only the better configuration *E_opt._* with higher *CF* is reserved for the next iteration *E_N +_*
_1_ (step **S-4**). It is noted that the lower *CF* can be reserved under a specific condition (e.g., *CF_N_* − *CF_N −_*
_1_ < −0.02) for the next iteration to jump out the local optima. The next configuration is perturbed by a random function ranging from −0.30 to 0.30 mA. In the level-2 adjustment, the current values can also be fixed after their random initialization. A linear scan is then applied to continue the search. For a more efficient search, the local current can jump between randomization and serial scan. Again, in the level-2 adjustment, each microheater has an independent value, and all four microheaters update at the same time. The process repeats until the *CF* values increase above a satisfaction threshold, e.g., 0.9. Each of the iteration results is saved to build a data library of the designed MMI. The following search can start with the thermal configuration with the best *CF* in the recorded library to improve efficiency. All the judgement conditions and search methods can be flexibly changed and revised to meet different targets. Details of the program can be found in the software Appendix A.

## 3. Results

Figure 7 shows the captured output spots for four different targets. The bar figures under each spot photo denote the comparison between the experimental and target values. Figure 7a–c show three different splitting powers under the lossless scenario. Figure 7d shows the lossy adjustment when a 3 dB attenuation is intended in the target total power. Further experiments show that all power splitting targets, lossless or lossy, can be optimized with *CF* > 0.9 within 300 iterations. More splitter configurations and their iteration processes can be found in the Appendix A.

Figure 8 shows the *CF* progress for the **[0.33:0.33:0.33]** case over the iterations, in which the device transforms from the initial 3 dB two-port splitter to an even three-port splitter. The *CF* rises to 0.969 after 166 iterations. The insets of Figure 8 are the captured photos of the output spots for the iteration No. 6, 90, 99, and 166, respectively. The heat power configurations of these four iterations are summarized in Table 1. The resistances of H_1_ to H_4_ at room temperature are measured as 163.8 Ω, 175.8 Ω, 176.1 Ω, and 163.1 Ω, respectively. The current (mA) configurations (H_1_ to H_4_) over the iterations are summarized in Figure 9. For comparison, a simple sweeping method without feedback mechanism, e.g., from 0 to 10 mA in step of 0.5 mA for all the four microheaters subsequently, leads to a total iteration number as high as 1.6×10^5^, while our program can already find the target settings within a few hundred iterations. The iteration process under the target **[0.33:0.33:0.33]** can be viewed in the Appendix A.

In Figure 8, the cost functions oscillate dramatically over the iterations. This is mainly caused by the coarse adjustment of level-1: when the adjacent iterations are both out of the target range, the reinitialized configurations cause a large difference on the output fields. When the present total output power is out of the target range while the last iteration is inside, large jumps can still happen, such as the iterations from No. 30 to 40. If the actual total power in consecutive iterations all stays within the target range, it shows a continuous gradient descent under the fine adjustment of level-2, such as the iterations from No. 61 to 78, in which the current values also change smoothly, as shown in Figure 9a–d.

It must be pointed out that the comparison between the actual and target total output power is necessary in each iteration. If the difference of the total output powers becomes large without being noticed, the iteration in the fine adjustment of level-2 can be easily trapped in the time-consuming loops within the local optima. It may also lead to wrong directions, e.g., results with the same relative splitting ratio but with different absolute splitting powers, such as **[0.33:0.33:0.33]** and **[0.10:0.10:0.10]**.

The coarse adjustment by level-1 brings a specific uncertainty that completely discards the previous configuration. Although this may increase the total number of the iterations, it brings robustness in finding better solutions beyond local optima. The simultaneous adjustments by the microheaters allow the target to be reached with possibly different configurations, i.e., the solution may not be unique. The vastly different heat power configurations between No. 90 (*CF_90_* = 0.846) and No. 166 (*CF_166_* = 0.969), shown in Table 1, have demonstrated the program’s ability to jump over locally optimized solutions, and the *CF* can be further improved along the increasing iterations.

## 4. Discussion

In our current technology, each iteration takes 1 s to complete, considering the response time of the camera, synchronization, and processing in MATLAB. There are a few ways to speed up the process. Firstly, high-speed PDs can be integrated on the FPWE instead of camera shot to record monitor the output power. The receiver response time can be shorted to sub-nanosecond. A tap coupler can be added to the input waveguide and divert part of light to a monitor diode for a more accurate reference to evaluate the lossless/lossy conditions. Thirdly, the thermo-optic tuning mechanism can be replaced by the ultrafast electro-optic (EO) effect using EO polymers [26] or by carrier injection method on junction semiconductor structures. The modulation time can also be shorted to sub-nanosecond scale. As only simple computation without matrix operation is performed in the optimization, the processing time for each iteration can be short. The computation takes roughly 0.1 millisecond on our lab PC (Intel Core i7-11700 @ 2.5GHz, 64 GB RAM, Windows 10) through the MATLAB program. The computation can be performed more efficiently by a customed MCU/FPGA system. With high-speed driver circuits integrated to transmit, receive, and process the data, we estimate the entire process to reach the target output can be improved to hundreds of microseconds for 300 iterations, which is comparable with the speed from the state-of-the-art MEMS-based switches.

The search program can also be improved in our future work, especially when developing a light flow controller with a large number of output ports and control electrodes. Firstly, the iteration configurations can be partially reserved with specific rules based on the target functions. For example, when *CF* already approaches a high value, a counter can be set so that the adjustment level-1 should be trigged less frequently in order to examine the local optima more closely. Secondly, each microheater can be applied with independent adjustment levels based on their contributions for different targets. For example, H_2_ and H_3_ may influence the light path to output Port2 more prominently than H_1_ and H_4_ because they are physically placed closer to the output port. In this case, the adjustment levels of H_2_ and H_3_ can be made finer to hit the sweet spot. The electrodes can be grouped based on their locations and contributions to the target. The comparison between the present and the previous values can be extended to a block of consecutive values, which can provide a balanced information for the next configuration.

Furthermore, an “offline” program can be developed as the first-round “coarse” search to limit the range of input variations so as to speed up the “online” search process. An equivalent AI-based neural network can be introduced to represent the MMI as the black box to map the input and output. To do so, a large dataset must be collected, followed by subsequent modelling and training. This equivalent network can then be solved reversely to find the closest input under a given output. In the end, the “online” search program can narrow in on the suggested values and perform the “fine” search. While the online search program is generic, the offline approach must be repeated each time the MMI structure or the electrode layout is changed. Depending on the specific applications, one may choose online, offline, or hybrid approach.

## 5. Conclusions

In conclusion, we have proposed and demonstrated a 1 × 3 lossless and lossy light flow controller with arbitrary splitting powers based on a programmable multimode waveguide engine. Taking into account the complex nature of thermal tuning on a multimode waveguide and further on the intermodal interference pattern, a search program with multi-level adjustment was developed to update the thermal current on the chip, record the optical feedback, and make adjustment accordingly until the target splitting power is reached. The implemented waveguide engine provides “online” control of the light flow on an optical chip under the passive constraint but with arbitrary attenuation. The experimental results showed a good agreement with the targets. The optimization program proves to be robust regardless of the initial conditions. The system can also be used to construct a look-up table to realize a large number of switching conditions without running the search program again. Time-consuming thermal and electromagnetic simulations are avoided. Further improvement to the system response time as well as to the search program are discussed. This work may bring a new concept on the real-time inverse design of programmable photonic devices for advanced switching and computing applications.

## Figures and Tables

**Figure 1 micromachines-13-01990-f001:**
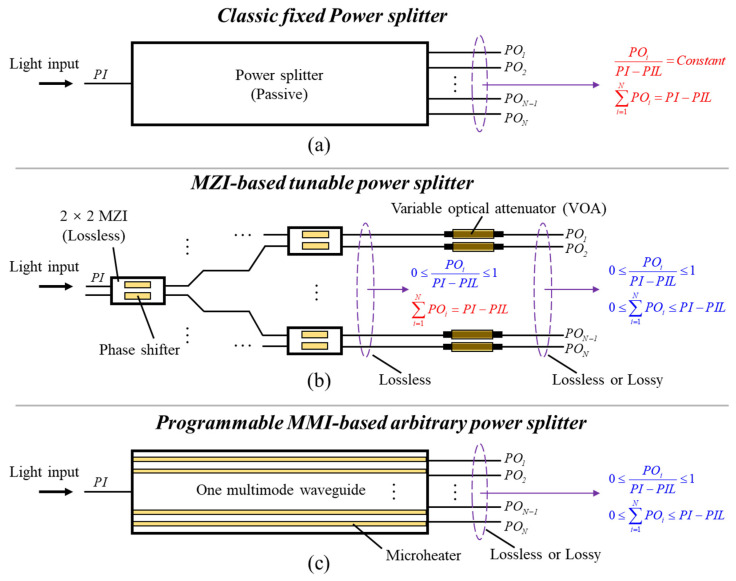
Schematic of the (**a**) classic fixed power splitter, (**b**) MZI-based tunable power splitter integrated with variable optical attenuators (VOAs), and (**c**) programmable MMI-based arbitrary power splitter. *PO_i_*, power of the output Port *i*; *PI*, power of the input light; *PIL*, lost power due to device insertion loss in practice.

**Figure 2 micromachines-13-01990-f002:**
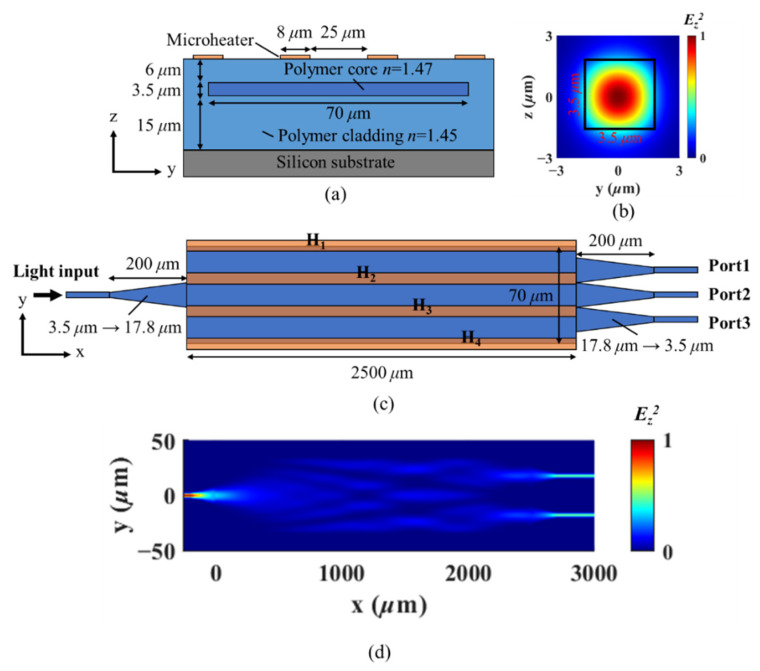
(**a**) Cross-section of the polymer multimode waveguide. (**b**) The field profile of the 3.5 × 3.5 μm^2^ single-mode waveguide. (**c**) Schematic of the 1 × 3 light flow controller. (**d**) The simulation results showing the device works as an even 3 dB splitter (Port1 and Port3) for the initial case without any heater power applied.

**Figure 3 micromachines-13-01990-f003:**
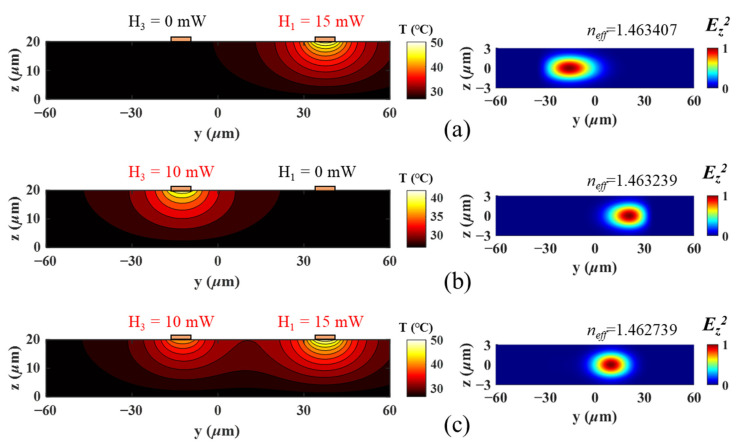
The temperature gradient and the field profile of the fundamental eigenmodes under different thermal configurations: (**a**) H_1_ = 15 mW, H_3_ = 0 mW; (**b**) H_1_ = 0 mW, H_3_ = 10 mW; and (**c**) H_1_ = 15 mW, H_3_ = 10 mW.

**Figure 4 micromachines-13-01990-f004:**
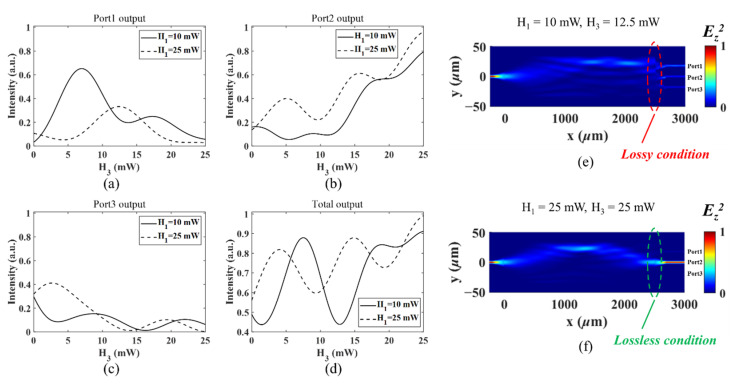
The simulated optical intensity variation of (**a**–**c**) Port1–Port3 and (**d**) their total outputs when H_3_ varies from 0 to 25 mW under different H_1_ values. The simulated results under different thermal configurations: (**e**) H_1_ = 10 mW, H_3_ = 12.5 mW (lossy condition); (**f**) H_1_ = 25 mW, H_3_ = 25 mW (lossless condition). All the intensity values are normalized by the total output intensity of the initial 3 dB splitter (Port1 and Port3) without any heater power applied.

**Figure 5 micromachines-13-01990-f005:**
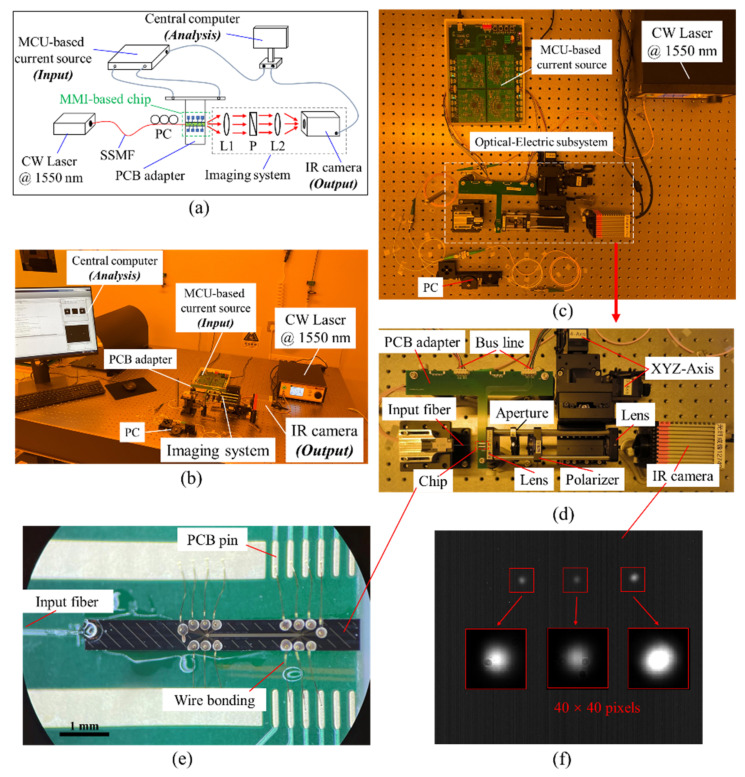
(**a**) The diagram and (**b**–**d**) the actual photos of the FPWE system. (**e**) The microscope photo of the optical chip on the adapter and with the input fiber. (**f**) Actual IR camera shot of the chip output facet. Inset shows the identified output spots within the 40 × 40 pixels (red squares). SSMF, standard single-mode fiber; PC, polarization controller; L1&L2, lens; P, polarizer.

**Figure 6 micromachines-13-01990-f006:**
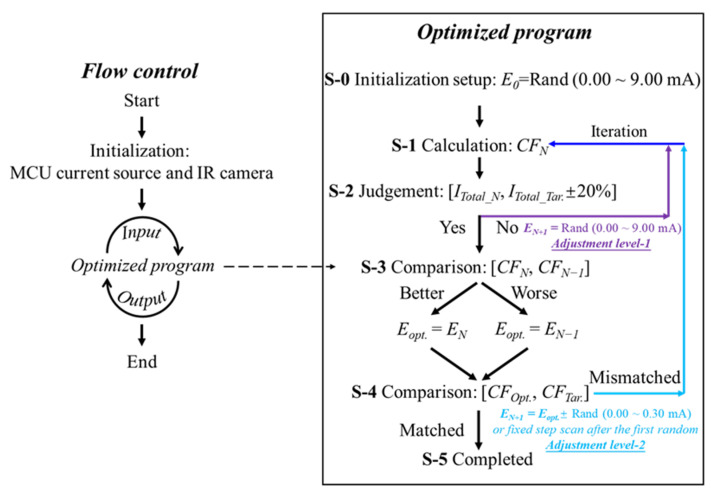
The diagram of the flow control and the multi-level search optimized program. *CF*, cost function; *E*, configuration of H_1_ to H_4_; Rand, random function; *I_Total_*, total output power of the three spots; *Opt.*, optimized; *Tar.*, targeted; *N* − 1, last iteration; *N*, present iteration; *N + 1*, next iteration.

**Figure 7 micromachines-13-01990-f007:**
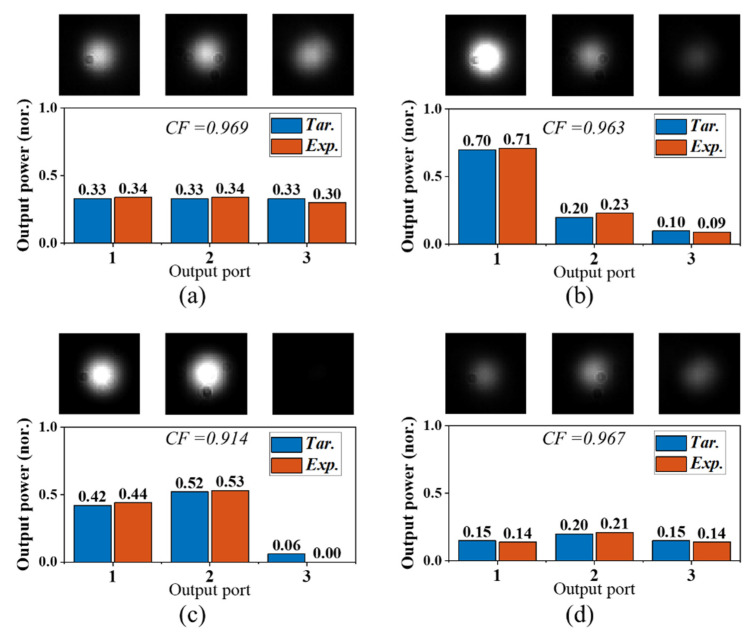
The experimental output spots and their comparisons with the targets of (**a**) **[0.33:0.33:0.33]**, (**b**) **[0.70:0.20:0.10]**, (**c**) **[0.42:0.52:0.06]**, and (**d**) **[0.15:0.20:0.15]**. The power values are normalized to the total output power at the initial state. The sum is 1 for lossless splitting, and the sum is <1 when extra attenuation is added.

**Figure 8 micromachines-13-01990-f008:**
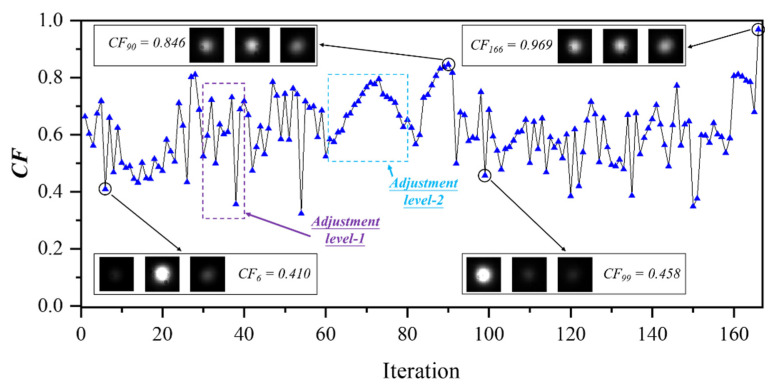
The relationship between the cost function and the iteration numbers for the case **[0.33:0.33:0.33]**. Insets are the actual shot photos of the intermediate results (No. 6, 90, 99) and the final result (No. 166).

**Figure 9 micromachines-13-01990-f009:**
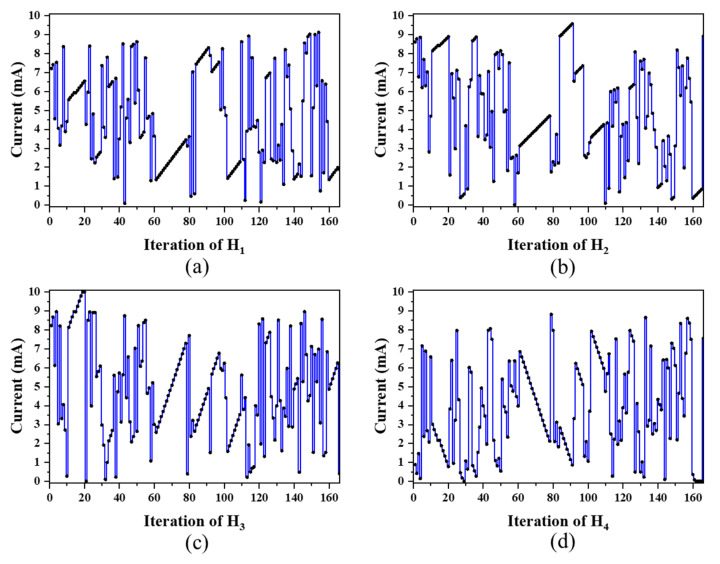
(**a**–**d**) The current configurations of H_1_ to H_4_ in the iteration.

**Table 1 micromachines-13-01990-t001:** Microheater power settings for different iteration (Unit: mW).

No.	H_1_	H_2_	H_3_	H_4_
6	1.6	10.4	11.8	0.9
90	11.0	15.8	3.8	0.2
99	11.2	1.1	6.0	0.7
166	0.6	14.0	0.0	9.3

## Data Availability

The data presented in this study are available from the corresponding author upon reasonable request.

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
