# Peer review of "Micro Light Flow Controller on a Programmable Waveguide Engine"

_micromachines, 2022, doi:10.3390/mi13111990_

Round 1

Reviewer 1 Report

The manuscript presents an interesting novel device developed through an original pragmatic approach, relying on experiments rather than simulations. Results and conclusions are convincing, but some key improvements and clarifications are mandatory before it can be considered for publication.

1. Fig. 1.b is not quite right, as there is no MZI. Either the rectangles represent MZIs themselves (but in this case the depicted phase shifters are pointless) if they are indeed 3dB couplers, they must be properly connected to form MZIs.

2. The authors claim that the thermal crosstalk makes it too difficult to treat the system analytically. This is actually not true. Recent publications (e.g., https://doi.org/10.1109/JLT.2019.2892512) have clearly shown that one can determine the thermal eigenmodes of the system and study it based on those orthogonal modes. Still, it is probably true that it is not immediate to determine the impact of those modes on a multimode waveguide, which justifies the experimental approach adopted by the authors.

3. The authors measured the excess loss compared to the case with all heaters off. I wonder how stable the light source is, especially when activating the heater can lead also to non-negligible back-reflections. I would have rather monitored the laser output with a tap-coupler and normalized the measurements relative to that power monitor.  The authors should comment on this potential issue. 

4. A couple of minor issues: 

- pag. 2, line 52: Please avoid using "etc." in a scientific paper. If you have in mind other implementations, just list them, otherwise place a full stop.

- pag. 6, line 191: "effected" should be "effect" instead.

Author Response

We thank the Reviewer for the comments. Please find our reply in the attached document.

Reviewer 2 Report

In the manuscript entitled "Micro Light Flow Controller on a Programmable Waveguide Engine", the authors study the 1x3 MMI-based programmable arbitrary power splitter, which requires fewer controls and has a smaller footprint than a conventional splitter on the chip. Even though the device response is no longer analytical and features cross-talk, the authors demonstrate an algorithm to find the target device response within hundreds of iterations. This transfers the optimization procedures from time-consuming simulation to real-time switching. Overall, this work enlightens an exciting topic on compact and efficient programmable PIC development. Therefore, I strongly recommend the publication of this work with minor modifications addressing the below comments:

1. What's the (relative) resolution and noise level of the IR camera pixel? 

2. The texts in Fig.8 need an exchange: iterations from #30 to #40 should be adjustment level-1 and iterations from #60 to #80 should be adjustment level-2

3. According to fig 9, during adjustment level 2, the heater current configurations seem like a scan, not a random search described in figure 6. Could the authors please explain?

4. In line 317, are the heater resistances measured when the heaters are nearly room temperature? What's the thermal coefficient of the heater when applying a relatively higher power?

5. The response of the heater is typically linear to heater power or square of the current. Randomization in power space (or I^2) might improve uniformity in different optimization scales.

6. Besides the conservative/dissipative nature of the system, are there any other limitations on the possible splitter configuration? How to choose the initial heater current limit to ensure the target could be found in the parameter space?

Author Response

(The authors gave the same response as above.)
